# VISA: Variational Inference with Sequential Sample-Average Approximations

**Heiko Zimmermann**
h.zimmermann@uva.nl

**Christian A. Naesseth**
c.a.naesseth@uva.nl

**Jan-Willem van de Meent**
j.w.vandemeent@uva.nl

Amsterdam Machine Learning Lab
University of Amsterdam
Amsterdam, The Netherlands

## Abstract

We present variational inference with sequential sample-average approximations (VISA), a method for approximate inference in computationally intensive models, such as those based on numerical simulations. VISA extends importance-weighted forward-KL variational inference (IWFVI) by employing a sequence of sample-average approximations, which are considered valid inside a trust region. This makes it possible to reuse model evaluations across multiple gradient steps, thereby reducing computational cost. We perform experiments on high-dimensional Gaussians, Lotka-Volterra dynamics, and a Pickover attractor. We demonstrate that VISA can achieve comparable approximation accuracy to standard importance-weighted forward-KL variational inference while requiering significantly fewer samples for conservatively chosen learning rates.

## 1 Introduction

Gradient-based methods have become the workhorse for inference in simulation-based models. When a model defines a fully differentiable density, methods based on Hamiltonian Monte Carlo (Duane et al., 1987; Neal et al., 2011) and reparameterized variational inference (Kingma & Welling, 2013; Rezende et al., 2014) are often considered the gold standard for generating high quality samples from the posterior distribution. However, it is not always practical or possible to use a differentiable model. The implementation of the simulator may not support differentiation, or the model itself may not be differentiable, for example because it employs discrete random variables or stochastic control flow. In such cases, inference falls back on methods based on score-function estimators (Glynn, 1990; Wingate & Weber, 2013; Ranganath et al., 2014) or IWFVI, which derives from reweighted wake-sleep methods (Bornschein & Bengio, 2015; Le et al., 2018). These methods are often less efficient, as a larger number of samples is required to compensate for higher variance gradient estimates. This can be problematic when a model is expensive to evaluate, for example because evaluation involves numerical simulations. Nonetheless, these methods remain the most viable option in a substantial number of use cases.

In this paper, we present VISA, a method that can substantially improve the computational efficiency of variational inference for models that are non-differentiable and computationally intensive. VISA is designed for applications where evaluation of the variational approximation is cheap relative to that of the model. Here we can save computation by reusing model evaluations across multiple updates of the variational posterior. To this end, we adapt IWFVI to employ a series of sample-average approximations (SAAs) (Nemirovski et al., 2009; Kim et al., 2015), which use a fixed set of samples

that defines a deterministic surrogate to the objective, rather than generating a fresh set of samples at each gradient step.

SAA-based methods were recently studied in the context of reparameterized black-box variational inference (Giordano et al., 2024; Burroni et al., 2024) which optimizes the reverse KL-divergence. These methods fix samples from a parameter-free distribution, which are transformed to samples from the approximate posterior using a differentiable map, whose parameters are optimized to maximize the variational bound. VISA differs from these methods in that it optimizes a forward-KL divergence and does not require a differentiable model. Concretely, VISA fixes samples from a parameterized variational distribution, rather than samples from a parameter-free distribution. Since the variational distribution will change during optimization, we construct a new SAA whenever the optimization procedure leaves a trust region, which we define in terms of the effective sample size. The result is a drop-in replacement for IWFVI that re-uses samples as much as possible, thereby saving computation.

We evaluate VISA in the context of three experiments. We first consider high-dimensional Gaussians, where the approximation error can be computed exactly. We then consider inference in a Lotka-Volterra system and a Pickover Attractor, where numerical integration is performed as part of the forward simulation. Our results show that VISA with a conservative (i.e. smaller than needed) step size can converge in a smaller number of model evaluations than IWFVI with a more carefully tuned step size. These results come with the caveat that VISA is more susceptible to bias than IWFVI, especially when used with a low effective sample size threshold. Our experiments show, that VISA converges substantially faster than IWFVI for conservartively chosen learning rates, while VISA performs on par with IWFVI for more carefully tuned step sizes.

## 2 Background

We first briefly review variational inference (VI) with stochastic gradient descent (SGD) and SAAs, before we introduce VISA in Section 3. Readers familiar with these topics can safely skip ahead.

### 2.1 Variational Inference

VI approximates an intractable target density with a tractable variational distribution by solving an optimization problem. The objective is typically to minimize a divergence measure $\mathcal{D}$ between the variational approximation $q_\phi$ with parameters $\phi \in \Phi$ and the target density $\pi$,

$$\min_{\phi \in \Phi} \{L(\phi) := \mathcal{D}(q_\phi, \pi)\}. \tag{1}$$

We assume that the target density is the posterior of a probabilistic model $\pi(z) = p(z \mid y)$ for which we are able to point-wise evaluate the joint density $p(y, z)$. The two most common approaches to VI are to minimize the reverse or forward KL divergence, for which objectives can be defined in terms of a lower bound $-L_\mathrm{R}$ and upper bound $L_\mathrm{F}$ on the log marginal $\log p(y)$,

$$L_\mathrm{R}(\phi) = - \mathbb{E}_{q_\phi}\left[\log \frac{p(y, z)}{q_\phi(z)}\right] \geq -\log p(y), \qquad L_\mathrm{F}(\phi) = \mathbb{E}_{p(\cdot|y)}\left[\log \frac{p(y, z)}{q_\phi(z)}\right] \geq \log p(y).$$

We will briefly discuss standard reparameterized VI, which maximizes the lower bound $-L_\mathrm{R}$, and IWFVI, which minimizes the upper bound $L_\mathrm{F}$.

**Reparameterized VI.** When maximizing a lower bound with stochastic gradient descent, we can either employ score-function estimators (Ranganath et al., 2014; Paisley et al., 2012), which tend to exhibit a high degree of variance, or make use of the reparameterization trick to compute pathwise gradient estimates (Kingma & Welling, 2013; Titsias & Lázaro-Gredilla, 2014; Rezende et al., 2014).

Reparameterization generates samples from a pushforward density $z = T_\phi(\xi) \sim q_\phi$ by fist sampling $\xi \sim q_\xi$ from a distribution independent of $\phi$ and then transforming the these samples using a differentiable function $T_\phi$. Under conditions that allow to exchange the order of differentiation and integration we can compute an unbiased estimate of the gradient

$$-\frac{d}{d\phi} L_\mathrm{R}(\phi) = \mathbb{E}_{q_\xi}\left[\frac{d}{d\phi} \log \frac{p(y, T_\phi(\xi))}{q_\phi(T_\phi(\xi))}\right] \approx \frac{1}{N} \sum_{i=1}^{N} \frac{d}{d\phi} \log \frac{p(y, T_\phi(\xi^{(i)}))}{q_\phi(T_\phi(\xi^{(i)}))}, \qquad \xi^{(i)} \sim q_\xi.$$

Importantly, reparameterized VI requires a model $p(y, z)$ that is differentiable with respect to $z$ in order to compute the pathwise derivative.

**Importance-Weighted Forward-KL VI.** Approximating the gradient of the forward KL-divergence does not require differentiability of the model $p(y, z)$ with respect to $z$, but does require approximate inference to generate samples from the posterior $p(z \mid y)$. IWFVI uses self-normalized importance sampling to propose samples from the variational distribution and reweights them according to the posterior by introducing an importance weights $\bar{w}_\phi = p(y, z)/q_\phi(z)$

$$-\frac{d}{d\phi} L_{\mathrm{F}}(\phi) = \mathop{\mathbb{E}}_{p(\cdot|y)}\left[\frac{d}{d\phi} \log q_\phi(z)\right] \simeq \sum_{i=1}^{N} \hat{w}_\phi^{(i)} \frac{d}{d\phi} \log q_\phi(z^{(i)}), \qquad \hat{w}_\phi^{(i)} = \frac{\bar{w}_\phi^{(i)}}{\sum_{j=1}^{N} \bar{w}_\phi^{(j)}},$$

where $z^{(i)} \sim q_\phi$. The resulting estimate is biased but consistent, meaning that it converges almost surely to the true gradient as $N \to \infty$.

## 2.2 VI with Sample-Average Approximations

Sample-average approximations (see Kim et al. (2015) for a review) approximate an expected loss with a surrogate loss in the form of a Monte Carlo estimate. In contrast to standard VI, the samples that the SAA is based on remain *fixed* throughout the optimization process. This means that the surrogate objective can be treated like any other deterministic function, which can be optimized using standard optimization tools. Concretely, a sample-average approximation applies to an optimization problem of the form

$$\min_{\phi \in \Phi}\left\{L(\phi) := \mathop{\mathbb{E}}_{\rho}\left[\ell(z, \phi)\right]\right\}, \tag{2}$$

in which the density $\rho(z)$ does not depend on the parameters $\phi$. This means that we can compute a surrogate loss $\hat{L}(\phi)$ that is an unbiased estimate of the original loss $L(\phi)$ by averaging over samples from $\rho$,

$$\hat{L}(\phi) = \frac{1}{N} \sum_{i=1}^{N} l(z^{(i)}, \phi), \qquad\qquad z^{(i)} \sim \rho.$$

Under mild conditions, as the number of samples $N \to \infty$, the minimizer $\hat{\phi} = \arg\min_\phi \hat{L}(\phi)$ and the minimal value $\hat{L}(\hat{\phi})$ converge almost surely to the minima $\phi^* = \arg\min L_\phi(\phi)$ and minimal value $L(\phi^*)$ of the original problem.

In the context of reparameterized VI, a sample-average approximation can be constructed by fixing a set of samples $\{\xi^{(i)} \sim q_\xi\}_{i=1}^{N}$ from a distribution independent of $\phi$,

$$\hat{L}_{\mathrm{R}}(\phi) = \frac{1}{N} \sum_{i=1}^{N} \log \frac{p\big(y, T_\phi(\xi^{(i)})\big)}{q_\phi\big(T_\phi(\xi^{(i)})\big)}.$$

In an SAA-based approach to reparameterized VI (Giordano et al., 2024; Burroni et al., 2024), optimization of the parameters $\phi$ will move the transformed samples $z^{(i)} = T_\phi(\xi^{(i)})$ to match the posterior density, whilst keeping the noise realizations $\xi^{(i)}$ fixed. Empirical evaluations show that combining the SAA approximation with an off-the-shelf second-order optimizer can result in substantial computational gains as well as more reliable convergence to the optimum.

## 3 SAA for Forward-KL Variational Inference

The primary motivation behind existing SAA-based methods for reparameterized VI (Giordano et al., 2024; Burroni et al., 2024) is that fixing the noise realizations defines a completely deterministic surrogate objective $\hat{L}_{\mathrm{R}}(\phi)$, which is compatible with standard second-order optimization methods. The main requirement from an implementation point of view is that the model density $p(y, z)$ is differentiable with respect to $z$. In this setting, it is also necessary to evaluate the model for every update, since any change to $\phi$ also changes the values of the transformed samples $z^{(i)} = T_\phi(\xi^{(i)})$.

**Algorithm 1** VISA

**Input:** Initial param. $\phi_0$, trust region threshold $\alpha$, data $y$

$\quad \tilde{\phi} \leftarrow \phi_0 \qquad\qquad\qquad\qquad\quad$ ▷ Initialize proposal parameter

$\quad \mathcal{Z} \leftarrow \{z^{(i)} \sim q_{\tilde{\phi}}\}_{i=1}^N \qquad\qquad$ ▷ Initialize samples

$\quad \hat{L}_\mathrm{F}(\phi \, ; \tilde{\phi}) = \sum_{i=1}^N \hat{w}_{\tilde{\phi}}^{(i)} \log \frac{p(y, z^{(i)})}{q_\phi(z^{(i)})} \qquad$ ▷ Initialize SAA

$\quad$ **for** $t = 1, \dots, T$ **do**

$\qquad \phi_t = \text{optimizer-step}(\hat{L}_\mathrm{F}, \phi_{t-1})$

$\qquad$ **if** $\phi_t \notin S_{\mathcal{Z},\alpha}(\tilde{\phi})$ **then** $\qquad$ ▷ Not inside trust region

$\qquad\quad \tilde{\phi} \leftarrow \phi_t \qquad\qquad\qquad\quad$ ▷ Update proposal

$\qquad\quad \mathcal{Z} \leftarrow \{z^{(i)} \sim q_{\tilde{\phi}}\}_{i=1}^N \qquad$ ▷ Refresh samples

$\qquad\quad \hat{L}_\mathrm{F}(\phi \, ; \tilde{\phi}) = \sum_{i=1}^N \hat{w}_{\tilde{\phi}}^{(i)} \log \frac{p(y, z^{(i)})}{q_\phi(z^{(i)})} \quad$ ▷ Refresh SAA

$\qquad$ **end if**

$\quad$ **end for**

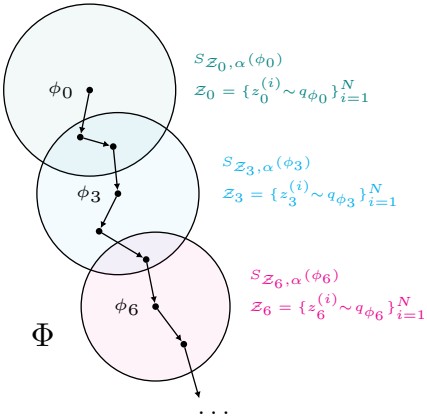

Figure 1: Visualization of parameter traces and trust regions corresponding to different SAAs. If after an update $\phi \notin S_{\mathcal{Z},\alpha(\tilde{\phi})}$, we set $\tilde{\phi} \leftarrow \phi$ to construct a new SAA and corresponding trust region.

In developing VISA, both the motivation and implementation requirements are somewhat different. Our primary interest is in minimizing the total number of model evaluations at convergence. We also wish to develop a method that is applicable when the model density $p(y, z)$ is not differentiable, either because the implementation simply does not support (automatic) derivatives, or because the model incorporates discrete variables or stochastic control flow, which introduce discontinuities in the density $p(y, z)$. To this end, we propose a method that optimizes a forward KL with an importance weighted objective that incorporates ideas from SAA-based approaches.

In a setting where we already have access to samples from the posterior, we can trivially define a SAA for the upper bound $L_\mathrm{F}(\phi)$,

$$\hat{L}_\mathrm{F}(\phi) = \frac{1}{N} \sum_{i=1}^N \log \frac{p(y, z^{(i)})}{q_\phi(z^{(i)})}, \qquad\qquad z^{(i)} \sim p(\cdot \mid y).$$

In practice, this naive approach is unlikely to be useful in a setting where evaluation of $p(y, z)$ is computationally expensive, since we would still need to carry out approximate inference to generate a set of samples from the posterior. We therefore adopt the approach used in IWFVI, which uses the variational distribution as a proposal in a self-normalized importance-sampler. To define an SAA for the objective in this setting, we express the objective at parameters $\phi$ in terms of an expectation with respect to a distribution from the same family with fixed parameters $\tilde{\phi}$. This allows us to approximate the objective by means of a sample-average approximation $\hat{L}_\mathrm{F}(\phi \, ; \tilde{\phi})$ with samples $z^{(i)} \sim q_{\tilde{\phi}}$,

$$L_\mathrm{F}(\phi) = \mathop{\mathbb{E}}_{p(\cdot|y)} \left[ \log \frac{p(y, z)}{q_\phi(z)} \right] = \mathop{\mathbb{E}}_{q_{\tilde{\phi}}} \left[ w_{\tilde{\phi}}(z) \log \frac{p(y, z)}{q_\phi(z)} \right] \approx \sum_{i=1}^N \hat{w}_{\tilde{\phi}}^{(i)} \log \frac{p(y, z^{(i)})}{q_\phi(z^{(i)})} =: \hat{L}_\mathrm{F}(\phi \, ; \tilde{\phi})$$

The quality of the approximation depends on how closely the proposal with parameters $\tilde{\phi}$ matches the posterior. Since our approximation of the posterior will improve during optimization, we will update $\tilde{\phi}$ to the current parameter values $\phi$ at some interval, resulting in an approach that we will refer to as a *sequential* SAA.

To determine when we need to generate a fresh SAA, we will introduce a notion of a trust region. This serves to define an optimization process in which the SAA is refreshed whenever the optimization trajectory leaves the current trust region. This optimization process is illustrated in Figure 1 and described schematically in Algorithm 1. We begin by setting the proposal parameters $\tilde{\phi} = \phi_0$ to the initial variational parameters, generating a set of samples $\mathcal{Z} = \{z^{(i)}\}_{i=1}^N$ and defining an SAA of the objective $\hat{L}_\mathrm{F}(\phi; \tilde{\phi})$ and a trust region $S_{\mathcal{Z},\alpha}(\tilde{\phi})$ based on these samples. We then repeatedly update $\phi_t$ using an optimizer until the value $\phi_t$ no longer lies in the trust region. At this point, we update the proposal parameters $\tilde{\phi} \leftarrow \phi_t$ and generate a fresh sample set, which we then use to update the SAA and the trust region.

**Defining Trust regions.** The importance weight in the objective can decomposed into two parts, (1) the ratio of the variational density and the trust region density, and (2) the ratio between posterior and variational density,

$$w_{\tilde{\phi}}(z) = v_{\phi,\tilde{\phi}}(z)\, w_\phi(z), \qquad\qquad v_{\phi,\tilde{\phi}}(z) = \frac{q_\phi(z)}{q_{\tilde{\phi}}(z)}.$$

The variance of $w_\phi$ is independent of the fixed proposal parameters and decreases as $q_\phi$ approaches the posterior $p(z \mid y)$ during optimization. Similarly, the variance of $v_{\phi,\tilde{\phi}}$ measures the mismatch between $q_{\tilde{\phi}}$ is to $q_\phi$, but can be set to zero by updating the proposal parameters to the parameters of the current variational approximation. The variance of the importance weights contributes to the variance of the gradient updates, which we generally expect to increase with increased variance of the weights. Accordingly, we want a trust-region criteria that allows us to reuse samples as long as the proposal distribution does not differ too much from the variational distribution, to limit the itroduction of additional variance and bias compared to IWFVI.

We formalize the notion of a trust region $S_{\mathcal{Z},\alpha}$ by defining a set-valued function which, for a given threshold $\alpha$ and samples $\mathcal{Z} = \{z^{(i)}\}_{i=1}^N$, maps each parameter to a corresponding trust region based on a scoring function $s_{\mathcal{Z}}$.

$$S_{\mathcal{Z},\alpha}(\tilde{\phi}) = \{\phi \in \Phi \mid s_{\mathcal{Z}}(\tilde{\phi},\phi) > \alpha\}.$$

In other words, for a given threshold $\alpha$ we can verify $\phi \in S_{\mathcal{Z},\alpha}(\tilde{\phi})$ by checking $s_{\mathcal{Z}}(\tilde{\phi},\phi) > \alpha$. Intuitively, the scoring function function should measure how well the proposal- and variational approximations, $q_{\tilde{\phi}}$ and $q_\phi$, match on a particular set of samples $\mathcal{Z}$. We visualize how new trust regions, corresponding to different SAAs, are constructed sequentially during optimization in Figure 1.

In this work, we propose to use the effective sample size (ESS) (Kong, 1992) as a trust-region criteria which can be motivated as a proxy measure of the variance of the importance weight or from the perspective of minimizing a $\chi^2$-divergence. More specifically, let $q_\phi$ be the current variational approximation and $q_{\tilde{\phi}}$ the current proposal, then the following relationships hold,

$$D_{\chi^2}(q_\phi \mid q_{\tilde{\phi}}) = \mathrm{Var}_{q_{\tilde{\phi}}}\left[\frac{q_\phi(z)}{q_{\tilde{\phi}}(z)}\right] \approx \frac{n}{n_{\mathrm{eff}}} - 1, \quad n_{\mathrm{eff}} = \frac{\left(\sum_{j=1}^N v_{\phi,\tilde{\phi}}(z_i)\right)^2}{\sum_{i=1}^N v_{\phi,\tilde{\phi}}(z_i)^2}, \quad v_{\phi,\tilde{\phi}}(z) = \frac{q_\phi(z)}{q_{\tilde{\phi}}(z)},$$

where $z_i \sim q_{\tilde{\phi}}$ (see Appendix A for a derivation). We see that a higher ESS indicates a lower $\chi^2$-divergence and lower variance of the importance weights. Consequently, we chose our scoring function

$$s_{\mathcal{Z}}(\tilde{\phi},\phi) = \frac{n_{\mathrm{eff}}(\mathcal{Z},\phi,\tilde{\phi})}{n}, \qquad\qquad n_{\mathrm{eff}}(\mathcal{Z},\phi,\tilde{\phi}) = \frac{\left(\sum_{z \in \mathcal{Z}} v_{\phi,\tilde{\phi}}(z)\right)^2}{\sum_{z \in \mathcal{Z}} v_{\phi,\tilde{\phi}}(z)^2},$$

where we express the ESS as a function of the sample set and distribution parameters, and normalize such that we can set the threshold parameter $\alpha \in [0,1]$.

Note that, assuming sampling from the proposal distribution is cheap, we could evaluate the ESS on a fresh set of samples from $q_{\tilde{\phi}}$, instead of using the sample set $\mathcal{Z}$. This would allow us to use a potentially much larger set of samples and hence obtain a better approximation of the variance of the importance weight. However, because we do not refresh samples at every iteration, we need to take into account that the actual sample set $\mathcal{Z}$ is not a perfect representation of the current proposal parameter $\tilde{\phi}$ (especially in the small-sample domain). As a consequence, we want our scoring function to not only represent how well $q_{\tilde{\phi}}$ matches $q_\phi$, but how well the distributions match for the particular set of samples $\mathcal{Z}$.

**Effect of the Trust-Region on Convergence.** The gradient estimator of VISA is expected to exhibit higher variance and bias than IWFVI as $q_\phi$ is generally closer to the posterior than $q_{\tilde{\phi}}$. As a result, decreasing $\alpha$ impedes the improvement of the gradient estimates of VISA compared to IWFVI as it delays updating the proposal distribution. As $\alpha \to 1$ the frequency of sample acquisition increases and, for $\alpha = 1$ VISA reduces to standard IWFVI.

For a low $\alpha$, the algorithm might converge to a locally optimal parameter $\hat{\phi}$ within the trust region of the current SAA that does not yet satisfy our global convergence criteria, i.e. minimizing the forward KL-divergence. In these cases, if the variational approximation is not degenerate, we can recover by increasing $\alpha$ such that $\hat{\phi} \notin S_\alpha(\tilde{\phi})$ and continue optimization. In this work we propose to choose $\alpha$ high enough that the algorithm does not converge prematurely to an optima of an intermediate SAA. In these cases we find that convergence of the training loss can be used as indicator for convergence of the forward KL-divergence or corresponding upper bound, which we verify in our experiments. We also experimented with caching past sample sets to compute a secondary loss based on the last $M$ SAAs. While we did not find it to add additional value in our experiments, it be a useful tool to assess convergence in other settings.

**Efficient Implementation.**    To avoid recomputing density values for old sample locations, we cache both sample location $z^{(i)}$ and the corresponding log-joint density of the model $\log p(z^{(i)}, y)$. If sampling from the proposal is cheap and memory is of concern, e.g. for large samples set or if past sample sets are stored to compute a validation loss, we can store the random seed instead of the sample and rematerialize the sample when needed.

# 4    Related Work

**VI with SAAs.**    While there exist several works that incorporate SAAs in VI (Gianniotis et al., 2016; Giordano et al., 2018; Wycoff et al., 2022), only recent work (Giordano et al., 2024; Burroni et al., 2024) has been focused on theoretically and empirically evaluating SAAs in the context of reparameterized black-box VI. These methods optimize a reverse KL-divergence and rely on reparameterization to move samples to areas of high posterior density while keeping a fixed set of noise realizations from the base distribution, which does not depend of the variational parameters. Optimizing the resulting deterministic objective allows the use of second-order optimization and linear response methods (Giordano et al., 2015) to fit covariances. This allows for substantial gains in terms of inference quality and efficiency but requires differentiability of the model.

**Stochastic second-order optimization.**    There is also work outside of the context of SAAs that aims to incorporate second order information to improve stochastic optimization and variational inference. Byrd et al. (2016) propose batched-L-BFGS, which computes stable curvature estimates by sub-sampled Hessian-vector products instead of computing gradient differences at every iteration. This work has been further adopted to the variational inference setting by Liu & Owen (2021). Pathfinder (Zhang et al., 2022) uses a quasi-Newton method to find the mode of the a target density and construct normal approximations to the density along the optimization path. The intermediate normal approximations are consequently used to define a variational approximation that minimizes an evidence lower bound. Similar to SAA-based methods, pathfinder can significantly reduce the number of model evaluations, but requires a differentiable model.

**VI and trust-region optimization.**    There are various works that combine VI with trust region optimization, e.g. with natural gradient descent (Theis & Hoffman, 2015), policy search (Arenz et al., 2018), or automatic differentiation VI to leverage second-order information within trust regions (Regier et al., 2017). However, in contrast to VISA, these methods do not optimize a forward KL-divergence and impose additional constraints on the variational distribution.

**VI with forward KL-divergence.**    VISA is also related to other methods that aim to optimize a forward KL-divergence or its stochastic upper bound. This includes reweighted-wake sleep (and wake-wake) methods (Bornschein & Bengio, 2015; Le et al., 2018) to which we compare VISA in the experiment section, as well as their doubly-reparameterized variants (Tucker et al., 2018; Finke & Thiery, 2019; Bauer & Mnih, 2021), which are not directly comparable as they require a differentiable model. While the methods above use a single importance sampling step using the variational approximation as a proposal, other methods use more complex proposal including MCMC proposals (Naesseth et al., 2020; Zhang et al., 2023), approximate Gibbs kernels (Wu et al., 2020), or proposal defined by probabilitic programs (Stites et al., 2021; Zimmermann et al., 2021). While these methods do not necessarily require a differentiable proposal they are not designed to be sample efficient but to approximate complex target densities.

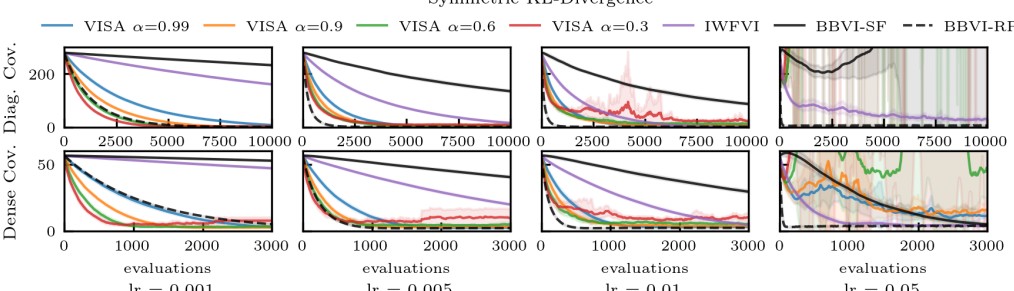

Figure 2: Symmetric KL-divergence as a function of the number of model evaluations for a Gaussian target with diagonal covariance matrix (top row) and dense covariance matrix (bottom row). For small learning rates (0.001, 0.005, 0.01) IWFVI and BBVI-SF, need a larger number of model evaluations to converge. VISA converges much faster as it compensates for the small step size by reusing samples. For a learning rate of 0.05 VISA becomes unstable and fails to reliably converge, while IWFVI still converges. For an even higher learning rate all methods but BBVI-RP fail to converge.

## 5 Experiments

We numerically evaluate VISA on three tasks, targeting a medium- to high- dimensional Gaussians, Lotka-Volterra dynamics, and a Pickover Attractor model. For all experiments we use Adam (Kingma & Ba, 2015) as an optimizer with the learning rates as indicated in the experiments. For the Gaussian experiment we report additional results with alternative optimizers in Appendix B.

### 5.1 Gaussians

To study the effect of different learning rates and ESS threshold parameters, we first evaluate VISA on approximating medium- to high- dimensional Gaussians and compare the inference performance over the number of model evaluations to IWFVI, standard reparameterized variational inference (BBVI-RP) and variational inference using a score-function gradient estimator (BBVI-SF). Notably, we include BBVI-RP as a reference only, showcasing that faster convergence can be achieved by leveraging the differentiability of the model. To allow for a fair comparison between methods that optimize a forward KL-divergence (VISA, IWFVI) and methods that optimize a reverse KL-divergence (BBVI), we evaluate the inference qualify in terms of the symmetric KL-divergence.

We study two target densities, a (1) $D = 128$ dimensional Gaussian with a diagonal covariance matrix $C_{\mathrm{diag}}$ and (2) $D = 32$ dimensional Gaussian with a dense covariance matrix $C_{\mathrm{dense}}$.

$$C_{\mathrm{diag}} = \mathrm{diag}\left(\left[\sigma_{\min} + (i-1) * \frac{\sigma_{\max} - \sigma_{\min}}{D-1}\right]_{i=1}^{D}\right), \quad C_{\mathrm{dense}} = \left(\frac{AA^T}{||AA^T||_{\mathrm{F}}} + 0.1\mathbb{I}\right),$$

with $\sigma_{\min} = 0.1$, $\sigma_{\max} = 1$, and $A_{ij} \sim \mathcal{U}(0,1)$. Figure 2 shows the results for different learning rates lr $\in \{0.001, 0.005, 0.01, 0.05\}$ and ESS thresholds $\alpha \in \{0.3, 0.6, 0.9, 0.99\}$. We compute gradient estimates for VISA, IWFVI and BBVI-SF with $N = 10$ samples, and gradient estimates for BBVI-RP using a single sample. Each setting is evaluated based on 10 independent runs.

We observe that VISA converges substantially faster than IWFVI and BBVI-SF at lower learning rates (0.001, 0.005, 0.01), both for the diagonal- and dense covariance matrix. The difference in the convergence rate becomes less pronounced as the learning rate increases. For large learning rates (0.05) VISA fail to converge reliably, while IWFVI still converges. For even higher learning rates all methods but BBVI-RP fail to converge. Moreover, we observe that VISA with small $\alpha$ converges faster in the early stages of training but can fail to converge fully in the later stages of training if the thresholds is too low (see $\alpha = 0.3$ in Figure 2). This is a result of underestimating posterior variance as samples are not refreshed frequently enough to prevent overfitting to high weight samples. Comparing convergence across learning rates and ESS thresholds, we observe that VISA with $\alpha > 0.3$ converges faster or at the same rate as IWFVI with the same or higher learning rate.

---

Python code for the experiments is available on `https://github.com/zmheiko/visa`

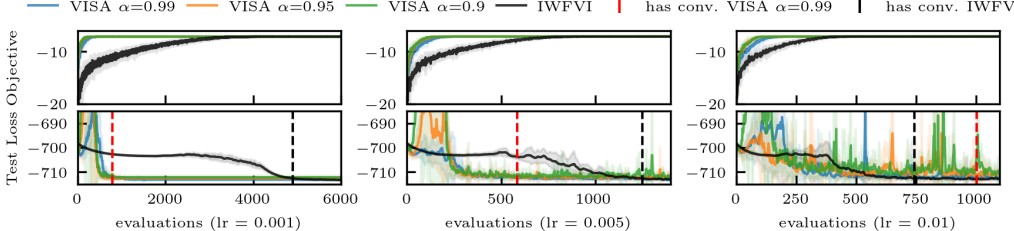

Figure 3: Results for Lotka-Volterra model with different learning rates. (Top row) Training objective over number of model evaluations. (Middle row) Approximate forward KL-divergence computed on reference samples obtained by MCMC. For smaller step sizes (0.001, 0.005) VISA achieves comparable forward KL-divergence to IWFVI while requiring significantly less model evaluations to converge (see vertical lines). For larger step sizes (0.01) VISA only converges with a high ess threshold (0.99) and requires more evaluations than IWFVI and VISA with a smaller step size (0.005).

## 5.2 Lotka-Volterra

The Lotka-Vorterra predator-prey population dynamics (Lotka, 1925; Volterra, 1927) are modeled by a pair of first-order ordinary differential equations (ODEs),

$$\frac{du}{dt} = (\alpha - \beta v)u, \qquad\qquad \frac{dv}{dt} = (-\gamma + \delta u)v,$$

where $v$ denotes the predator- and $u$ denotes the prey population. We will in the following denote the pair of predator-prey populations at time $t$ with $z_t = (u_t, v_t)$. The dynamics of the ODE are governed by its population growth and shrinkage parameters $\theta = (\alpha, \beta, \gamma, \delta)$, which we want to infer together with the initial conditions of the system given noisy observations $y_{1:T} = (y_1, \ldots, y_T)$. Following Carpenter (2018), we place priors over the initial population sizes $z_0$ and system parameters,

$$z_0^{\text{prey}}, z_0^{\text{pred}} \sim \text{LogNormal}(\log(10), 1), \quad \alpha, \gamma \sim \text{Normal}(1, 0.5), \quad \beta, \delta \sim \text{Normal}(0.05, 0.05),$$

and assume a fractional observation error,

$$y_t^{\text{prey}}, y_t^{\text{pred}} \sim \text{LogNormal}(\log z_t, \sigma_t), \qquad\qquad \sigma_t \sim \text{LogNormal}(-1, 1).$$

Given an initial population $z_0$, system parameters $\theta$, and observations $y_{1:T}$ we can solve the ODE numerically to obtain approximate population sizes $z_{1:T}$ for time steps $1, \ldots, T$ which we use to compute the likelihood of the observed predator-prey populations, $p(y_{1:T} \mid z_0, \theta) = \prod_{t=0}^{T} p(y_t \mid z_t)$.

Our goal is to learn an approximation to the posterior $p(\theta, z_0 \mid y)$ by minimizing the upper bound

$$L_{\text{F}}(\phi) := \mathbb{E}_{(z_0, \theta) \sim p(\cdot, \cdot \mid y)} \left[ \log \frac{p(z_0, y, \theta)}{q_\phi(z_0, \theta)} \right]. \tag{3}$$

We model the variational approximation $q_\phi$ for the interaction parameters $\theta$ and initial population sizes $z_0$ as jointly log-normal and initialize $\phi$ such that the the marginal over $z_0$ matches the prior and the marginal over $\theta$ has similar coverage to the prior. We approximate the objective and its gradient using $N = 100$ samples. To specify a common convergence criteria, we compute the highest common test loss value for VISA with $\alpha = 0.99$ and IWFVI that is not exceeded by more than 1 nat by all consecutive test loss values. The convergence threshold computed this way is $-712.7$ nats.

To evaluate the inference performance, we first generate $N = 100\,000$ approximate posterior samples using a No-U-Turn Sampler (NUTS) (Hoffman & Gelman, 2014) with $100\,000$ burn-in steps and window-adaption, which generally provides good performance out of the box. We use the approximate posterior samples to approximate an "oracle" for the upper bound in Equation (3),

$$\hat{L}_{\text{F}}^{\text{NUTS}}(\phi) = \frac{1}{N} \sum_{i=1}^{N} \log \frac{p(z_0^{(i)}, y, \theta^{(i)})}{q_\phi(z_0^{(i)}, \theta^{(i)})},$$

---

NUTS is an adaptive Hamiltonian Monte Carlo sampler and uses the gradient information of the model to guide the generation of proposals. As such it requires the log-joint density model to be differentiable which is not required by VISA or IWFVI.

which we evaluate along with the training objective during optimization to assess convergence.

We find that VISA is able to obtain variational distribution of similar quality to IWFVI while requiring significantly fewer model evaluations for smaller learning rates (see Figure 3). Interestingly, IWFVI requires significantly fewer model evaluations per gradient step during the early stages of training, while requiring slightly more evaluations per gradient step thereafter. We hypothesise that this is again a result of under approximating posterior variance in the later stages of training. As a result, even small changes in the variational distribution can lead to big changes in the ESS, which triggers the drawing of fresh samples. For IWFVI, we also find a more pronounced difference between different ESS thresholds and their influence on convergence. Runs with a higher ESS thresholds converge more stably and are able to achieve lower test loss in the final stages of training. The reported means and standard are computed based on 10 independent runs.

## 5.3 Pickover Attractor

Following Rainforth et al. (2016), we model a 3D Pickover attractor (Pickover, 1995) with system parameters $\theta = (\beta, \eta)$ (where $x_t \in \mathbb{R}^3$ is a vector and the superscript denotes the component),

$$x_{t+1}^1 = \sin(\beta x_t^2) - \cos(2.5 x_t^1) x_t^3, \quad x_{t+1}^2 = \sin(1.5 x_t^1) x_t^3 - \cos(\eta x_t^2), \quad x_{t+1}^3 = \sin(x_t^1).$$

Due to its chaotic nature the system is sensitive to small perturbations in its initial state, i.e. even small variations in the initial state lead to exponentially fast diverging trajectories. Therefore, to track the evolution of the system, we employ a bootstrap particle filter (Gordon et al., 1993) which assumes noisy observations $y := y_{1:T}$ and introduces auxiliary variables $z := z_{1:T}$ to model the latent state of the system. We define the prior over system parameters

$$p(\theta) = \begin{cases} 1/18 & -3 \leq \theta_1 \leq 3, 0 \leq \eta \leq 3 \\ 0 & \text{otherwise} \end{cases},$$

and model the transition and observation as,

$$z_0 \sim \mathcal{N}(\cdot \mid \mathbf{0}_3, \mathbb{I}_3), \qquad z_t \sim \mathcal{N}(\cdot \mid h(z_{t-1}, \theta), \sigma_z), \qquad y_t \sim \mathcal{N}(\cdot \mid z_t, \sigma_y),$$

where $\sigma_z = 0.01$, $\sigma_y = 0.2$, and $h$ evolves the system by one time step using the equations of the Pickover attractor described above. The particle filter is used to simulate $T = 100$ time steps with $M = 500$ particles, which renders evaluating the model expensive. To restrict the proposal to the same domain as the prior by first sampling from a Normal, which we denote $\bar{q}_\phi$ and parameterize by a mean and lower Cholesky factor, and then transform the sample by

$$f(\theta) = (\tanh(3 \cdot \theta_1), \tanh(1.5 \cdot \theta_2 + 1.5)).$$

The density of the transformed samples is $q_\phi(\theta) = \bar{q}_\phi(\theta) |\det \frac{df(\theta)}{d\theta}|^{-1}$, which is appropriately restricted to the domain of the prior.

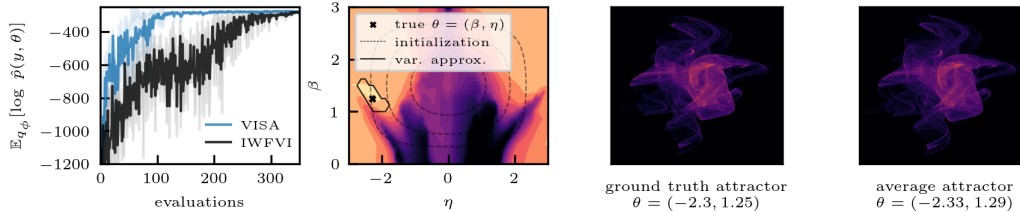

Figure 4: Results for Pickover attractor. (a) Approximate log-joint density over number of batch-evaluations of model. (b) Log-joint approximation plotted over domain of prior. The variational approximation capture the high density area containing the data. (c) Visualization of pickover attractor with ground truth parameters $\theta = [-2.3, 1.25]$. (d) Visualization of attractor with average system parameters computed over 10.000 samples from the learned variational approximation. Each evaluation in the plot corresponds to evaluating a batch of $N = 10$ samples.

We are interested in approximating the marginal posterior $p(\theta \mid y)$ over system parameters by optimizing the evidence upper bound

$$\underset{p_\theta(\theta|y)}{\mathbb{E}} \left[ \log \frac{p_\theta(y, \theta)}{q_\phi(\theta)} \right] = \underset{p_\theta(\theta|y)}{\mathbb{E}} \left[ \log \frac{\mathbb{E}_{p_{\mathrm{pf}}}[\hat{p}(y \mid \theta)]}{q_\phi(\theta)} \right] \leq \underset{p_\theta(\theta|y)}{\mathbb{E}} \left[ \mathbb{E}_{p_{\mathrm{pf}}} \left[ \log \frac{\hat{p}(y \mid \theta)}{q_\phi(\theta)} \right] \right].$$

To obtain a tractable objective we replace the intractable marginal likelihood $p(y \mid \theta) \approx \hat{p}_\theta(y \mid \theta)$ with the marginal likelihood estimate obtained by running the particle filter (Naesseth et al., 2019), similar to pseudo-marginal methods (Andrieu et al., 2010) and approximate the gradient with $N = 10$ samples. As the likelihood estimate is non-differentiable due to the discrete ancestor choices made inside the particle filter, we cannot run NUTS to obtain approximate posterior sample as before, but instead report the log-joint density of the variational distribution.

We observe that VISA converges more stably with fewer samples compared to IWFVI and find that attractors corresponding to samples from the variational approximation look qualitatively similar to those based on the true parameters. We summarize the result in Figure 4.

## 6  Discussion and Limitations

In this paper we developed VISA, a method for approximate inference for expensive to evaluate models that optimizes the forward KL-divergence through a sequence of sample-average approximations. Each SAA is optimized deterministically and fixes a single set of samples, hereby requiring new model evaluations only when the SAA is refreshed. To track the approximation quality of the current SAA, VISA computes the ESS of the ratio between the current variational distribution and the proposal distribution that was used to construct the SAA. If the ESS falls below a predefined threshold, a new SAA approximation is constructed based on fresh samples from the current variational distribution. We observe a significant reduction of required model evaluations for conservatively chosen step sizes, while achieving similar posterior approximation accuracy as IWFVI, the equivalent method that does not employ the sequential sample-average approximation. In the following we are discussing the limitations of our method.

**Underapproximation of posterior variance.**  Both reparameterized VI, which optimizes the reverse KL-divergence, and IWFVI, which optimizes the forward KL-divergence via importance sampling, are prone to under approximating posterior variance. In the case of reparameterized VI, this can often be attributed to the mode seeking behaviour of the reverse KL-divergence, while in IWFVI the low effective samples sizes can lead to over fitting to a small number of high-weight samples. We found that keeping the samples fixed for too long, i.e. using an ESS threshold that is too low, can exacerbate this problem, as the optimizer can take multiple steps towards the same high-weight samples.

Giordano et al. (2024) and Burroni et al. (2024) showed that when applying SAA to reparameterized VI, it is possible to make use of second-order methods. We experimented with optimizing SAAs with L-BGFS, which is a quasi-Newton method with line search. However, we found that in the setting of optimizing a forward KL with relatively few samples, L-BGFS can amplify the problem of overfitting, often leading to instabilities and collapsed variational distributions.

**Number of latent variables and parameters.**  Because VISA employs a relatively small number of samples and does not refresh samples at every iteration, we found that it is not well-suited to models with a large number of latent variables or large number of parameters. This agrees with theoretical findings by Giordano et al. (2024), who show that SAAs for a full covariance Gaussian fail if the number of samples is not at least in the same regime as the number of latent dimensions. Burroni et al. (2024) manage to train full covariance normal approximation by using a sequence of SAAs using increasingly large sample sizes, however, this is directly opposed to our goal of reducing the number of model evaluations in expensive to evaluate models.

## Impact Statement

This paper presents work whose goal is to advance the field of Machine Learning. While we recognize that the advancement of machine learning methods can have broad societal impact, we do not foresee specific consequences of this particular research.

## Acknowledgement

The authors would like to thank Tamara Broderick for a helpful discussion about sample-average approximations in the context of reparameterized variational inference.

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

## A    Motivation of trust region.

Using the ESS as a trust-region criteria can be motivated from a perspective of minimizing a $\chi^2$-divergence or the variance of the corresponding importance weight. Let $q$ be the current variational approximation and $\tilde{q}$ the current proposal, then the following relationship holds,

$$D_{\chi^2}(q \mid \tilde{q}) = \mathrm{Var}_{\tilde{q}}\left[\frac{q(z)}{\tilde{q}(z)}\right] = \frac{n}{n_{\text{eff}}} - 1, \qquad n_{\text{eff}} = \frac{\left(\sum_{j=1}^n w_j\right)^2}{\sum_{i=1}^n w_i^2}.$$

Note that the ESS criteria defined in section 3 uses the *normalized ESS* and hence differs from $n_{\text{eff}}$ by a factor $n$,

$$\frac{n}{n_{\text{eff}}} - 1 = \frac{1}{s_{\mathcal{Z}}(\tilde{q}, q)} - 1. \tag{4}$$

*Proof.*

$$D_{\chi^2}(q \mid \tilde{q}) = \int_{\mathcal{Z}} dz \, \frac{(q(z) - \tilde{q}(z))^2}{\tilde{q}(z)} = \int_{\mathcal{Z}} dz \, \frac{q(z)^2}{\tilde{q}(z)} - \underbrace{2\int_{\mathcal{Z}} dz \, \frac{q(z)\tilde{q}(z)}{\tilde{q}(z)} - \int_{\mathcal{Z}} dz \, \frac{\tilde{q}(z)^2}{\tilde{q}(z)}}_{=1}$$

$$= \mathbb{E}_q\left[\frac{q(z)}{\tilde{q}(z)}\right] - 1 = \mathbb{E}_{\tilde{q}}\left[\left(\frac{q(z)}{\tilde{q}(z)}\right)^2\right] - 1$$

$$\mathrm{Var}_{\tilde{q}}\left[\frac{q(z)}{\tilde{q}(z)}\right] = \mathbb{E}_{\tilde{q}}\left[\left(\frac{q(z)}{\tilde{q}(z)}\right)^2\right] - \underbrace{\mathbb{E}_{\tilde{q}}\left[\frac{q(z)}{\tilde{q}(z)}\right]^2}_{=1} = \mathbb{E}_{\tilde{q}}\left[\left(\frac{q(z)}{\tilde{q}(z)}\right)^2\right] - 1$$

$$\mathbb{E}_{\tilde{q}}\left[\left(\frac{q(z)}{\tilde{q}(z)}\right)^2\right] - 1 = \frac{\mathbb{E}_{\tilde{q}}\left[\left(\frac{q(z)}{\tilde{q}(z)}\right)^2\right]}{\mathbb{E}_{\tilde{q}}\left[\frac{q(z)}{\tilde{q}(z)}\right]^2} - 1 \approx \underbrace{\frac{n\sum_{i=1}^n w_i^2}{\left(\sum_{j=1}^n w_j\right)^2}}_{\text{Kish design effect}} - 1 = \frac{n}{n_{\text{eff}}} - 1.$$

$\square$

## B    Additional results for Gaussians

We evaluate the performance of VISA with two additional optimizers, RMSProp and SGD. We use the implementation and standard parameters provided by Optax. We find that VISA converges faster with Adam and RMSProp compared to SGD. We also observe that, for a small learning rate, SGD can get stuck in local optima, while RMSProp gets unstable more quickly with higher learning rates.

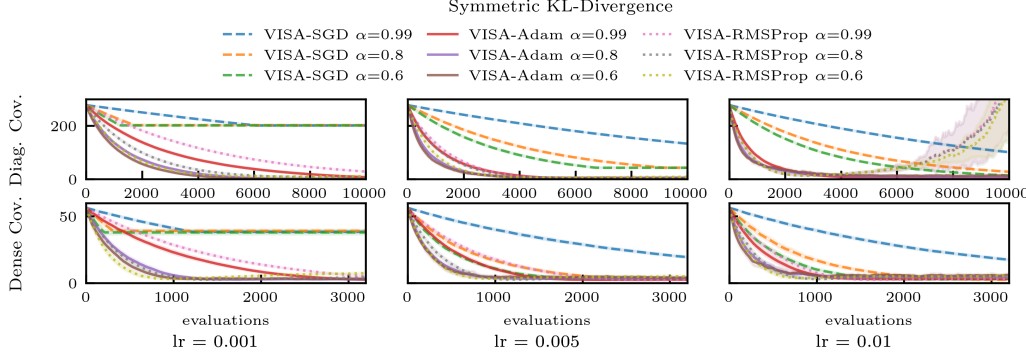

