# OpenReview forum: "VISA: Variational Inference with Sequential Sample-Average Approximations"
_NeurIPS.cc/2024/Conference — NeurIPS 2024 poster_

### Official Review · Reviewer_sEay · 2024-06-23

**Soundness:** 3
**Presentation:** 3
**Contribution:** 2
**Rating:** 5
**Confidence:** 3

**Summary:**

This paper presents a method for approximate inference in expensive to evaluate models where the gradients may not be available. The presented approach proceeds in a trust-region-optimization fashion where a series of deterministic surrogates objectives are optimized. Each objective is optimized until a trust region condition is met, indicating that the objective needs to refreshed. The proposed approach shows promising results when compared to other competing approaches.

**Strengths:**

The paper presents a simple procedure that is clearly laid out. It is well written for most parts and seems to be promising direction for future research.

**Weaknesses:**

- I think the use of ESS for determining the trust region is rather arbitrary. I did not find a sound explanation for why this should be the goto way for determining the trust region.
- Also, the analysis around the choice of alpha seems rather experimental. The performance seems extremely critical to the choice of this parameter and there doesn't seem to be any good way to select this parameter.
- Moreover, rerunning the optimization to select this parameter can result in a lot of evaluations. Since we wanted to avoid this in the first place, would a user not be better of using the IWFVI approach?
- Also, what exactly is the difference between the IWFVI approach and the standard Re-weighted wake sleep? Why rename an already named algorithm?
- I did not find enough details about the exact optimization procedure used to optimize the VISA approach. Which optimizer was used?
- I am also not sure how are the number of evaluations calculated for the Figure 2. Are we only counting the evaluations during the refresh step? If so, should we not see step decays in the metrics for VISA approaches?

**Questions:**

Please, see the Weaknesses section.

**Limitations:**

The scalability of the proposed approach is rather limited as pointed out by the authors. The proposed method seems extremely sensitive to the choice of learning rate and the alpha. It is unclear how to chose them properly.

---

> ### Author Rebuttal · Authors · 2024-08-06
>
> **Motivation for ESS as trust-region criteria.**
> See general response.
>
> **Choice of ESS threshold.**
> See general response.
>
> **IWFVI vs Reweighted Wake-Sleep (RWS).**
> RWS is used in the context of amortized variational inference and optimizes both the parameters of the model and the parameters of the (amortized) variational approximation. Moreover standard RWS optimizes different objectives to update the variational approximation for the sleep- and wake-phases of the algorithm, which use simulated and real data for training respectively. IWFVI only optimize the variational distribution and does not use simulated data from the model (sleep-phase in RWS). Hence, IWFVI updates correspond to the wake-phase updates to the variational parameters in RWS only; both are based on optimizing an importance-sampling based estimate of the forward KL-divergence.
>
> **Details on exact optimization procedure.**
> We use Adam with standard parameters (as implemented in optax) for all experiments. We will clarify this in the final version of the manuscript.
>
> **Number of evaluations.**
>  It is correct that only freshly drawn samples are counted towards the number of evaluations, hence the number of evaluations only increases after a “refresh step”. The reason why there is no visible staircase pattern in the corresponding plots is that VISA refreshes after a relatively small amount of evaluations (for conservatively chosen ess thresholds) and that the plots are averaged over multiple runs. For a single VISA run with a low enough ess threshold we would indeed see a visible staircase pattern.

---

> ### Comment · Reviewer_sEay · 2024-08-10
> **A few more clarifications.**
>
> Thank you for the response. I had a few more clarifying questions.
>
> 1. Can you provide numbers on how many times the samples are refreshed? At what stages are samples refreshed more? Is it during the start of the optimization or towards the end? Also, can you give a version of the plots with number of optimization iterations on the x-axis? I understand the main motivation is to reduce the number of model evaluations; however, I still think answer to these might improve my understanding of the method.
>
> 2. Is N the same across the experiments for both VISA and IWFVI? How is the N value selected? I wonder what are the trade-offs when selecting the N. It is likely that for a given setting of N, IWFVI suffers because it uses more samples at every update iteration.
>
> 3. From what I understand, the combined choice of $\alpha$ and learning-rate is tricky. Things work fine in the simple Gaussian examples; however, on real problems, it sounds like as a practitioner I will have to play around with different settings and see what works. This makes me squeamish--the ultimate aim of the setting of this paper is to reduce the number of model evaluations. As such, the method seems a bit understudied to me. Can the authors offer some comments on this thought process?
>
> Thank you.

---

> > ### Author Response · Authors · 2024-08-12
> > **Re: A few more clarifications.**
> >
> > Thank you for your questions and your interest in our methodology! We hope our response provides further clarification on the points you have raised.
> >
> > **Number of model evaluations.** We have prepared the requested plots illustrating inference performance over the number of optimization iterations, as well as additional plots showing the relationship between optimization iterations and model evaluations. However, after reviewing the guidelines for the discussion process, we realized that we are not permitted to include anonymous links to external sources. Consequently, while we will include the plots in the final manuscript, we must address the questions in this response without the aid of visual representations.
> >
> > In the plots in our experiment section, the number of times VISA refreshes samples corresponds directly to the number of evaluations. For instance, in the experiments with the full-covariance Gaussian, VISA with $\alpha \leq 0.9$ converges after approximately 1,000 model evaluations when using a learning rate of 1e-3. However, at this point, VISA has already performed around 30,000 gradient steps (optimization iterations). In contrast, for the same scenario, IWFKL also converges after about 30,000 gradient steps but requires 30,000 model evaluations, as it draws fresh samples at each iteration.
> >
> > Our results also show, across experiments, that VISA resamples at relatively constant intervals and uses slightly less model evaluations early in training. This makes sense, considering that the posterior approximation, and consequently the proposal distribution, is initialized relatively broad and the ESS criteria is not very sensitive to perturbations of the variational approximation within the coverage of the proposal. As training progresses, the variational approximation, and eventually the proposal, become more tightly peaked, causing the ESS criteria to become more sensitive to perturbations of the variational distribution, which might move it outside the coverage of the proposal. As a result VISA refreshes samples more frequently after the early phases of training.
> >
> > **Number of samples per model evaluation.** In our experiments, both VISA and IWFVI use the same number of samples per (batch) model evaluation, denoted $N$, which we heuristically selected to be sufficiently high to ensure stable convergence for IWFVI. In our experience, this approach works well as long as the ESS threshold is chosen conservatively. If $N$ were significantly increased, we would indeed expect to observe a greater difference in the overall number of model evaluations between IWFVI and VISA. Moreover, with a larger sample size, it would likely be possible to select a lower ESS threshold for VISA without compromising training stability, potentially leading to faster convergence. Conversely, if we chose $N$ to be the (unknown) minimal number of samples required for IWFVI to achieve stable convergence, then VISA, for $\alpha < 1$, would potentially suffer from instabilities and might not be able to reduce the number of model evaluations substantially. However, in practice, the minimal number of samples required is difficult to identify and might vary for different phases of training. In practical scenarios we expect VISA to be able to reduce the amount of model evaluations.
> >
> > **Tuning hyperparameters.** We acknowledge that introducing an additional hyperparameter can make practitioners hesitant, and often for good reason. However, it's important to note that IWFVI is a special case of VISA, for $\alpha=1$. Therefore, it is reasonable to base the hyperparameter selection for VISA on those used for IWFVI, as long as we choose a conservative ESS threshold, for which we expect VISA to behave similar to IWFVI. In our experiments, we applied this method for both the initial selection of the learning rate and the number of samples per model evaluation. While practitioners may choose to invest computational resources in optimizing $\alpha$ further, in many cases, VISA already outperforms IWFVI with the same hyperparameters and a conservative ESS threshold, e.g. $\alpha=0.99$.
> >
> > We believe VISA is particularly valuable for practitioners who currently use IWFVI for inference on models that are costly to evaluate. In such scenarios, VISA can serve as a drop-in replacement that requires minimal tuning. However, even when hyperparameters cannot be directly "bootstrapped" from previous experiments, we argue that finding appropriate hyperparameters for VISA is no more challenging than it is for IWFVI.

---

### Official Review · Reviewer_Bkjs · 2024-07-10

**Soundness:** 3
**Presentation:** 3
**Contribution:** 3
**Rating:** 6
**Confidence:** 3

**Summary:**

This paper introduces a new method for variational inference called VISA, which stands for Variational Inference Using Sequential Sample-Average Approximations. VISA is based on importance-weighted forward-KL variational inference and allows for reusing model evaluations across multiple gradient steps. This makes it suitable for non-differentiable and computationally intensive models. The authors conducted experiments using simulated and real data to validate VISA, showing that it can achieve similar approximation accuracy to standard IWFVI while requiring significantly fewer samples.

**Strengths:**

1. The paper is well-written and organized, making complex concepts easy to understand through a logical flow of information.
2. VISA effectively addresses the challenge of minimizing the number of model evaluations, thereby achieving significant computational efficiency.
3. The experiments conducted provide robust evidence that VISA converges faster, requiring fewer model evaluations compared to existing methods.

**Weaknesses:**

1. While the concept of trust regions based on Effective Sample Size (ESS) is intuitive, the paper would benefit from a more formal theoretical justification for the specific choice of trust region.
2. The paper currently lacks a discussion on the robustness of the proposed method to various optimization methods. While it touches on the use of L-BGFS, including an analysis of how VISA performs with other optimization techniques such as classical stochastic gradient descent (SGD) and adaptive methods like RMSProp and Adam would enhance the comprehensiveness of the study.

**Questions:**

1.	Could the algorithm's performance be enhanced by incorporating adaptive adjustments for the ESS threshold $\alpha$ and the learning rate?
2.	It appears there may be a typo in the caption for Figure 3 when referring to the methods VISA and IWFVI. Could you please verify and correct this if necessary to ensure clarity and accuracy in the presentation of your results?

**Limitations:**

The authors have adequately discussed the limitations of their approach.

---

> ### Author Rebuttal · Authors · 2024-08-06
>
> **Motivation for ESS as trust-region criteria.**
> See general response.
>
> **Robustness to optimization methods.**
> We indeed use Adam (optax implementation) for all experiments and will clarify this in the final version of the manuscript. Based on your suggestion, we conducted additional experiments for the Gaussian target densities to demonstrate that VISA can be used with SGD, Adam, and RMSProp for appropriately selected stepsizes. Our evaluations shows that plain SGD (without momentum or Nesterov acceleration) might run into local optima more easily as Adam and RMSProp for small step sizes, while RMSProp might not be as robust to larger step sizes as Adam. Overall, the experiment suggests that Adam is a good standard choice for VISA. We have included the corresponding convergence plots in the PDF file attached to the general rebuttal and will add a polished version to the final manuscript.
>
> **Adaptive ESS threshold.**
> While we did not test adaptive strategie, we experimented with using a schedule for the ess threshold. However, we did not find consistent improvement over using a fixed threshold and hence opted for the simplest possible presentation of the algorithm. That being said, we expect that problem specific scheduling and adaptation strategies are able to outperform fixed ess threshold. However, determining a performant schedule or adaption hyperparameters might require running inference multiple times, which contributes to the overall number of evaluations and complicates a fair comparison.
>
> **Typo in Figure 3.**
> There was indeed a typo, thanks for pointing it out! The caption should read: "For smaller step sizes (0.001, 0.005) VISA achieves
> comparable forward KL-divergence to IWFVI while requiring significantly less model evaluations
> to converge (see vertical lines). For larger step sizes (0.01) VISA only converges with a high ess
> threshold (0.99) and requires more evaluations than IWFVI and VISA with a smaller step size (0.005)."

---

> > ### Comment · Reviewer_Bkjs · 2024-08-12
> >
> > Thank you for your response, which clarified my concerns. I will maintain my current score.

---

### Official Review · Reviewer_kZyU · 2024-07-14

**Soundness:** 3
**Presentation:** 3
**Contribution:** 2
**Rating:** 6
**Confidence:** 3

**Summary:**

The paper proposes to use a sample-average approximation for variational inference. In contrast to previous works, the method uses the forward-KL which does not require differentiability of the joint likelihood. In order to sample from an approximate posterior instead of the exact posterior, a sequential trust-region approach is introduced. The method is evaluated on small-scale problems and some benefits of the forward-KL SAA approach compared to BBVI or reverse-KL SAA are demonstrated.

**Strengths:**

- The derivatinos and mathematical formulation are clear, the paper is easy to read.
- The proposed trust-region approach is rigorously evaluated on small-scale examples with various ablation studies.

**Weaknesses:**

- The proposed VISA method (and other baselines as well) seem quite sensitive to the learning rate and other tuning parameters. Therefore it is hard to judge whether the proposed approach has significant benefits over other methods, or whether the benefits are due to a tuning of the learning rate.
- The relevance of the considered applications to the wider machine-learning audience seems unclear. The work is motivated by non-differentiability but then only toy examples are shown. Some real-world applications (for example, reinforcement learning) would make the paper stronger.

**Questions:**

- When compared to other SAA methods (Giordano et al 2024), (Burroni et al 2023), the novelty of the proposed method seems to be in the use of forward KL rather than reverse KL.  I can see the computational advantages of using forward-KL, but does the mode-covering property of forward-KL pose problems sometimes? For example, when the approximate posterior is simple but the true posterior is multimodal with high loss inbetween?
- Natural-gradient VI methods can also be used in black-box settings -- see for instance "Variational Adaptive-Newton Method for Explorative Learning", Khan et al., 2017. Related natural evolution strategies (such as CMA-ES) have shown state-of-the-art performance in optimizing loss functions with only gradient evaluations. Have you considered comparing to these methods, or would you expect them to perform similary as BBVI-SF?

**Limitations:**

Limitations are adequately discussed in the last section of the paper.

---

> ### Author Rebuttal · Authors · 2024-08-06
>
> **Sensitivity to the learning rate.**
> See general response.
>
> **No real-world experiments.**
> See general response.
>
> **Failure cases of the Forward KL-divergence.**
> While we consider the mode-covering behavior of the forward KL-divergence as a positive, there can be instances where a mode of the variational approximation covers multiple modes of the target density and as a result assigns not insignificant mass to a low density area. To prevent these scenarios it is important to choose an appropriate variational approximation that allows to model the expected degree of multimodality. Moreover, in practice, optimizing an importance-weighted approximation to the forward KL-divergence often results in an under approximation of posterior variance similar to optimizing a reverse KL-divergence, especially when targeting well separated modes in the small sample regime.
>
> **Adding more specialized baselines.**
> We chose to compare VISA to IWFV and BBVI-SF because these methods are equally back-box, in the sense that they do not assume differentiability of the model and are not restrictive in the variational families that can be used. However, in settings where more specialized methods can be applied, we expect these methods to outperform our baselines and potentially VISA. One instance of this can be seen in the first experiment where we included a BBVI-RP baseline, which is generally considered a black-box method but leverages the differentiability of the model.
>
> The suggested baseline, VAN and CMA-ES, specifically use a multivariate Gaussian as a variational approximation and VAN-D makes use of a reparameterization trick to approximate the diagonal of the Hessian, which requires a differentiable model. That being said, we agree that it would be insightful to compare to and potentially combine VISA with a black-box (in the sense described above)  stochastic second-order method. We will work to include an appropriate method in the final manuscript.

---

> > ### Comment · Reviewer_kZyU · 2024-08-13
> > **Thanks for the clarifications!**
> >
> > I have carefully read the rebuttal, and some of my concerns were addressed. Other issues still stand (e.g., problems with forward KL), but I have concluded that this may just be an inherent limitation of the method which is not fixable.  I will increase my score accordingly.
> >
> > Regarding the real-word experiments, I am still not fully convinced that the method will be that useful in practice and unsure how it compares to other baselines (e.g., natural-gradient methods). But as the method is fully "black-box", releasing the algorithm as a software package to the community an interesting real world use case for it may emerge.

---

### Official Review · Reviewer_snbt · 2024-07-14

**Soundness:** 3
**Presentation:** 2
**Contribution:** 2
**Rating:** 4
**Confidence:** 3

**Summary:**

The authors propose to reduce the number of model evaluations during the optimization of the variational lower bound. To this end, they integrate sample avarage approximations (SAAs) into IWFVI framework, which allows updates of variational parameters while keeping approximate samples generated from a variational distribution obtained a few steps before during optimization. The method was evaluated on a few synthetic experiments.

**Strengths:**

1. The method is effective in the experiments considered (it requires less model evaluations to converge).
2. The clarity of the paper is good and is easy to follow.

**Weaknesses:**

1. The method seems to be a combination of a few engineering tricks. It is unclear what the effect of inconsistency between approximate samples ($\tilde{\phi}$) and variational distribution ($\phi$) during optimization is.
2. Only synthetic experiments are considered.
3. In the paper, the model is assumed to have no tubable parameters, while many applications requires tunable model as well (e.g. VAE).

**Questions:**

See weakness.

**Limitations:**

The authors thinks there is no social impact. However, no justifications are given.

---

> ### Author Rebuttal · Authors · 2024-08-06
>
> **Clarification regarding our methodology.**
> VISA does not optimize the reverse KL-divergence or corresponding variational lower bound but a forward KL-divergence or corresponding variational upper bound. This is a crucial difference to previous related work, which studies SAAs in the context of the reparameterized reverse KL-divergence.
>
> **Presentation.**
> Given the clarity of our paper was highlighted as a strength, we are slightly confused by the low score for the presentation. Please let us know in which ways we can further improve our presentation.
>
> **Inconsistency between proposal and variational distribution.**
> The discrepancy between the proposal and variational distribution is captured by the importance weight $q(z \mid \phi) / q(z \mid \tilde \phi)$, which contributes to the overall importance weight used in the sample average approximation and corrects for the error introduced by not sampling directly from $q(z \mid \phi)$. In general, the bias and variance of the gradient estimates will be higher if the proposal has not been updated recently and the discrepancy to the variational distribution is large. To quantify the effect that this discrepancy has on the overall performance of the algorithm numerically we conducted experiments with various ESS thresholds $\alpha$, which are a way to formalize an upper bound on the maximum allowed discrepancy between the proposal and variational distribution.
>
> **Only synthetic experiments.**
> See general response.
>
> **No tunable model parameters.**
> VAEs are trained by jointly optimizing a generative model and a variational posterior approximation by maximizing an ELBO in the context of amortized variational inference. In contrast, our work considers posterior inference for a given model, a setting that is frequently encountered when working with simulation-based models, and treats parameter estimation of that model as an orthogonal problem.

---

> > ### Author Response · Authors · 2024-08-12
> > **Request for clarification on remaining concerns**
> >
> > Dear Reviewer,
> >
> > We noticed that you provided a very low score, and we would like to know if our rebuttal has sufficiently addressed your concerns. If any issues remain unresolved, we would greatly appreciate it if you could share them with us at your earliest convenience, so we can address them in time.
> >
> > Thank you for your time and consideration!

---

> > ### Comment · Reviewer_snbt · 2024-08-13
> >
> > Thanks for your rebuttal, which clarifies part of my concerns (regarding inconsistency between proposal & variational distribution; no tunable model parameters), and I am willing to raise my score to 4 to reflect that. However, the lack of real-world experiments makes me hesitate to recommend this work for acceptance.

---

### Author Rebuttal · Authors · 2024-08-06

We would like to thank the reviewers for their detailed reviews! We are delighted to see the clarity of our manuscript and rigor of our experiments listed among the strengths of our work. We address the point raised by multiple reviewers below and respond to the individual concerns and questions in dedicated rebuttals under the corresponding reviews.

**Sensitivity to learning rate and choice of ESS threshold (kZyU, sEay).**
We believe that one of the strengths of VISA is its robustness to the choices of learning rate for conservatively chosen thresholds $\alpha$. As seen in the first two experiments, the rate of convergence of VISA is relatively stable across all but the largest presented learning rates, which are chosen too large on purpose. In contrast, the convergence rates of IWFVI and BBVI are more sensitive to the learning rate and only competitive when chosen close to the upper limit in terms of training stability. In general, while finding the optimal training hyperparameters for either method might require careful tuning, we found that a simple recipe for selecting the learning rate and ess-threshold for VISA is often enough to outperform IWFVI:
1) Choose the same learning rate as for IWFVI and a conservative ess-threshold, e.g. $\alpha=0.99$.
2) If convergence is stable, one might choose to reduce $\alpha$ to achieve even faster convergence, otherwise decrease the learning rate slightly.


**Motivation for ESS as a trust-region criteria (Bkjs, sEay).**
There is in fact a good reason to use the ESS to define the trust region, and we will clarify this in the manuscript. The Kish effective sample size is directly related to the variance of the importance ratio between the current variational distribution and the trust region distribution, which can in turn be expressed as a Chi-Square divergence between these distributions.

$
\chi^2(q(\cdot; \phi) \mid q(\cdot; \tilde{\phi})) =
 \mathrm{Var}_{q(\cdot; \tilde{\phi})} \left[\frac{q(z; \phi)}{q(z; \tilde{\phi})}\right]
\approx \frac{n}{\mathrm{ess}} - 1
$

As a result, the ESS-based trust region can be motivated from (1) a divergence perspective. i.e. we want the sampling distribution close to the variational distribution as measured by the chi-square divergence, or from (2) the perspective of the variance of the corresponding importance weight (or ESS as a proxy measure), which we want to be low (or high ESS respectively). The second perspective also motivates the use of the ESS to assess the quality of importance samplers or adaptive resampling strategies based on an ESS criteria.

We will expand upon the motivation of the effective sample size as a trust-region criteria in the manuscript and will add deviations for the above equalities to the appendix.

**Only synthetic experiments (snbt, kZyU).**
We see the contribution of this work in proposing VISA and studying its behavior by exploring a wide spectrum of hyperparameters. While this setting makes it somewhat challenging to evaluate VISA on truly expensive real-world models, we want to highlight that the Lotka-Volterra model and Pickover attractor capture the properties of many real-world scientific simulation models. The Pickover attractor model in particular can exhibit extremely chaotic behavior, is not fully differentiable, and is fairly expensive to evaluate. To be able to track the attractor state we run SMC with 500 particles for 100 time steps for each of the 100 samples that make up a single batch evaluation of the model.

**VISA runs with different optimizers (Bkjs).**
Please find convergence plots in the attached PDF file.

---

### Decision · Program_Chairs · 2024-09-25

**Decision:**

Accept (poster)

**Comment:**

This is a borderline paper.

Given an expensive-to-evaluate $p(y, z)$, not-differentiable wrt $z$ and known $y$, the goal of this paper is to speed up importance-weighted forward-KL variational inference (IWFVI), which optimizes a variational posterior without requiring differentiability wrt $z$.

This work stands on top two previously existing works:
   1. VI with Sample-Average Approximations: These use Reverse KL + SAA. But they require differentiability of $p(y, z)$ wrt $z$.
   2. Importance-weighted forward-KL VI (IWFVI): Does not require differentiability wrt $z$, but requires many samples of $p(y, z)$ to reduce the variance of the gradients.

The novelty of this work is the introduction of a mechanism to speed up IWFVI by reusing the samples of $p(y, z)$ as the optimization of the variational distribution $q(z)$ proceeds. $p(y, z)$ is evaluated at locations sampled from an initial $q(z)$. As we optimize $q$, if it doesn't change much, optimizations steps can proceed without re-evaluating the expensive $p(y, z)$. Whenever $q$ departs too much (above a threshold) from the last sampled one, a resampling is triggered and reevaluation of  $p(y, z)$ is triggered.

This idea accelerates IWFVI, by reducing the number of $p(y, z)$ evaluations required for a given performance.

On the negative side, this paper only includes synthetic experiments, and I agree with the critical reviewers that this paper would be strengthened by the inclusion of real-world use cases where $p(y, z)$ is very expensive to evaluate. Fortunately, some of the selected synthetic experiments are not trivial and do stress-test the proposed model, so it is reasonable to expect the conclusions of the current work to carry over to at least some real-world use cases.

Given this and the potential value of the novel algorithm, I rule in favor of acceptance.

Authors: T
- Take into account the reviewer's comments to strengthen the paper and improve its clarity.
- Clarify the use of ESS to define the trust region and include other points of contention raised in the discussion.
- Include a real-world experiment if possible. This can be taken from prior work on Forward-KL Variational Inference, such as Le et al. "Revisiting Reweighted Wake-Sleep". It should be easy to show a reduced number of evaluations to achieve the same performance.

Minor comments:

Under line 81, there's a sign error. Additionally, a $\phi$ is missing in $\hat{w}^{(i)}$.

Line 161 "we can recover by decreasing α" -> Increasing α.